# Biomechanical Stability and Osteogenesis in a Tibial Bone Defect Treated by Autologous Ovine Cord Blood Cells—A Pilot Study

**DOI:** 10.3390/molecules24020295

**Published:** 2019-01-15

**Authors:** Monika Herten, Christoph Zilkens, Fritz Thorey, Tjark Tassemeier, Sabine Lensing-Höhn, Johannes C. Fischer, Martin Sager, Rüdiger Krauspe, Marcus Jäger

**Affiliations:** 1Department of Orthopedics and Trauma Surgery, University of Duisburg-Essen, 45147 Essen, Germany; Monika.Herten@uk-essen.de (M.H.); Tjark.Tassemeier@uk-essen.de (T.T.); 2Orthopedic Department, Heinrich-Heine-University Düsseldorf, 40225 Düsseldorf, Germany; Christoph.zilkens@med.uni-duesseldorf.de (C.Z.); Sabine.Lensing-Hoehn@med.uni-duesseldorf.de (S.L.-H.); Ruediger.Krauspe@med.uni-duesseldorf.de (R.K.); 3Center for Hip, Knee and Foot Surgery, Sports Traumatology Department, ATOS Hospital, 69115 Heidelberg, Germany; Fritz.Thorey@atos.de; 4Institute for Transplantation Diagnostics and Cell Therapeutics, Heinrich-Heine-University Düsseldorf, 40225 Düsseldorf, Germany; Johannes.Fischer@med.uni-duesseldorf.de; 5Animal Research Institute, Heinrich-Heine-University Düsseldorf, 40225 Düsseldorf, Germany; Martin.Sager@med.uni-duesseldorf.de

**Keywords:** umbilical cord blood cells (USSC), hydroxyapatite, bone regeneration, critical size defect, external fixateur, rigidity measuring device, tibia, sheep

## Abstract

The aim of this study was to elucidate the impact of autologous umbilical cord blood cells (USSC) on bone regeneration and biomechanical stability in an ovine tibial bone defect. Ovine USSC were harvested and characterized. After 12 months, full-size 2.0 cm mid-diaphyseal bone defects were created and stabilized by an external fixateur containing a rigidity measuring device. Defects were filled with (i) autologous USSC on hydroxyapatite (HA) scaffold (test group), (ii) HA scaffold without cells (HA group), or (iii) left empty (control group). Biomechanical measures, standardized X-rays, and systemic response controls were performed regularly. After six months, bone regeneration was evaluated histomorphometrically and labeled USSC were tracked. In all groups, the torsion distance decreased over time, and radiographies showed comparable bone regeneration. The area of newly formed bone was 82.5 ± 5.5% in the control compared to 59.2 ± 13.0% in the test and 48.6 ± 2.9% in the HA group. Labeled cells could be detected in lymph nodes, liver and pancreas without any signs of tumor formation. Although biomechanical stability was reached earliest in the test group with autologous USSC on HA scaffold, the density of newly formed bone was superior in the control group without any bovine HA.

## 1. Introduction

Extensive bone defects pose a considerable challenge in orthopaedic surgery. Particularly, bone lesions above a critical size become scarred rather than regenerated, leading to non-union [1]. Autologous bone grafts are currently the clinical gold standard with their osteogenic, osteoconductive, and osteoinductive qualities [2]. The resources for autografts are limited and associated with several problems, including infection risk and donor site morbidity [3]. In contrast, application of bone substitutes such as allografts or natural or synthetic biomaterials, are only osteoconductive and show a lack of osteoinductivity, resulting in insufficient callus formation and incomplete bridging when applied to a critical size bone defect [4]. However, these substances act as scaffolds, providing an osteoconductive framework for new bone formation. Thus, biomaterials and allografts are clinically applied to small defects. In larger defects they are usually combined with osteoinductive stimuli such as autologous bone, growth factors, bone marrow concentrate, or platelet rich plasma (PRP) for promoting the migration, proliferation, and differentiation of bone cells.

Bone marrow stromal cells (BMSC) have an especially strong potential for clinical application since they stimulate bone healing in large segmental defects, compared with synthetic void fillers alone [5,6,7,8,9,10]. Moreover, it is evident that undifferentiated progenitor cells may have less immunogenic potency than fully differentiated cells [11,12,13,14,15].

Besides bone marrow derived stromal cells, less differentiated mesenchymal stem cells from umbilical cord (unrestricted somatic stem cells, USSC) are a promising candidate for tissue regeneration [16,17]. USSC are multipotent and can differentiate into cells of all three germ layer lineages: endodermal (liver, lung) [18,19], ectodermal (nerve) and mesodermal (heart, cartilage, bone, fat and blood) [20,21]. They show a high osteoregenerative potential in vitro [22,23,24] and a low immunogenic profile compared to other adult stem cell types [25]. Due to their immaturity, USSC express HLA class I at low levels and are negative for MHC class II, and they are therefore particularly attractive to manipulate or modify graft-versus-host disease (GVHD) [26]. It was demonstrated that MHC-mismatched cord blood cells did not induce a detectable immune response in an animal model [27]. Some data indicate that cord blood MSC have a direct immunosuppressive effect on proliferation of T lymphocytes from human adult peripheral blood (PB) and umbilical cord blood (UCB) in vitro [28]. 

We hypothesize that mesenchymal progenitor cells from the umbilical cord display an osteogenic differentiation potential in vivo and are capable of regenerating critical size osseous defects. Autologous, labeled USSC were applied to an ovine tibial bone size model and investigated as to bone regeneration. The novelty of the present study is the use of autologous USSC for bone repair with the concurrent monitoring of bone regeneration by radiography and biomechanical stability. These experiments include a new approach in non-embryonic stem cell research with the potency for clinical implementation.

## 2. Results

### 2.1. Characterization of Autologous USSC

The mesenchymal stem cell character of the cells was confirmed: USSC proliferated efficiently and reached confluency after 8 to 10 days, FACS analysis displayed a negative signal for CD14, CD34, and CD45 and a positive signal for CD44 and CD90. The differentiation into the three lines was demonstrated in Figure 1.

### 2.2. In Vitro Assessment of the USSC-HA Scaffold

The ovine USSCs seeded onto the HA-scaffolds were present on the outer surface of the biomaterial after seven days in vivo (Figure 2). At further follow-up, cell viability testing revealed that the cells completely covered the outer surface of the scaffold but were also detected at deeper levels up to 1 mm from the surface.

### 2.3. Labeling Control

When the cells were allowed to remain confluent, fluorescence was maintained. Cell passaging and therefore cell proliferation was associated with a reduced fluorescence. The PKH-26 labeling was determined by flow cytometry: cell viability as measured by live/dead labeling was above 95%. The fluorescent labeling did not affect the survival and proliferation of cells in vitro as could be demonstrated in unchanged cell viability.

### 2.4. In Vivo Assessment of Bone Healing

Within a few hours the animals recovered successfully from surgery and remained in good health. The operated tibias were stably fixed with the external fixateurs except for one animal from the HA group, which had a complete dislocation of the fixateur and was euthanized one week after surgery. Temporary pin infections occurred once within each group during the six-month follow-up period. The animals used the operated leg postoperatively without any dysfunction and showed weight bearing on the extremity. However, during the six-month follow-up period, muscle wastage was observed on the operated limb in all sheep suggesting an abnormal weight bearing pattern. No signs of infection, inflammation, or immunogenic reactions could be detected at any time.

### 2.5. Biomechanical Measurements

Biomechanical testing presented a torsion distance of 3.8 ± 1.8 mm at surgery decreasing to 1.2 ± 0.3 mm after one month in the test group (Figure 3). At two, three, four, and five months, the torsion force threshold of 20 N was reached immediately with 0 mm torsion distance. In the HA group, torsion distance was 2.8 mm at surgery, 1.8, 0.9, and 1.1 mm after one, two, and three months, respectively. At four and five months, the threshold was reached immediately. In the control group the stiffness distance was 3.0 ± 0.1 mm after surgery and 1.8 ± 1.1, 1.8 ± 2.0, 1.7 ± 2.3; 1.2 ± 1.6 after one, two, three, and four months respectively. At five months, the threshold was reached immediately.

### 2.6. X-Ray Analysis

The images of the bone defect area revealed comparable radiographic signs of profound new bone formation and bridging of the defect site after six months (Figure 4). Increased callus formation could be observed in the test group after two, three, and four months, which completely disappeared after five months.

### 2.7. Histological Results of Bone Formation

Representative histological sections of the healed bone defects are displayed in Figure 5. While mature bone was observed throughout the defect with lamellar aspect and small lacunae in the test group (Figure 5a,d) and in the control group (Figure 5b,e), immature bone was observed with larger lacunae and disorganized mineralized matrix in the HA group (Figure 5c). The distance between the original bone fragments varied from 16.8 to 22.7 mm, resulting in a measurement area of the former tibial bone defect of 298 to 457 mm^2^ (Table 1). Histomorphometric analysis revealed that the area of newly formed bone was largest in the control group with 82.5 ± 5.5% compared to 59.2 ± 13.0% in the test group and 48.6 ± 2.9% in the HA group. Bone regeneration was superior in the control group without any bovine hydroxyapatite (Table 1).

### 2.8. Homing of the USSC

Membrane-labeled USSC could be detected in lymph nodes, spleen and liver of the test group animals without any signs of tissue alterations (Figure 6).

## 3. Discussion

In our study we described an autologous ‘large-animal model’ which is useful in evaluating bone regeneration from cord blood. Moreover, we confirmed previously published data that ovine cord blood cells (ovine USSCs) are able to differentiate into osteo-, chondro-, and adipoblasts. It was demonstrated that the applied HA scaffold allowed new bone formation, and the cells migrated into the micropores to a measurable extent. However, the control group which was treated without any biomaterial showed bone healing as well.

### 3.1. Subcritical Nature of the Defect

In the empty control defect, homogenous bone formation and complete bridging were observed after the healing period of six months, indicating a subcritical nature of the defect. Critical-sized defects are defined as the smallest size intraosseous wound in a particular bone and species of animal that will not heal spontaneously during the lifetime of the animal. In the present study a defect size of 2 cm was chosen, based on previous reports that a 1.8 cm [29] and 2.0 cm [30] defect was the smallest critical-sized defect described in sheep tibiae. However, in other ovine bone defect studies, larger tibial defect size such as 3.0 [31], 3.5 [32], or 5.0 cm [33] were used without leading to union after three months. Apart from size, a critical defect is influenced by several factors such as age, health status/comorbidities, anatomical location, fixation technique, biomechanical conditions, and of course healing period. Based on our data, we assume that a tibial bone defect of 2 cm is not critical for the Bentheimer sheep breed [34].

### 3.2. Less Bone Occurred in the Scaffold Groups

The HA scaffold application did not perform as well as the empty defect in bone healing. From orthopedic and trauma surgery we know that bone substitutes such as bioceramics are not only promoting bone healing in non-critical size bone defects but may delay bone formation and act as a stress raiser. Due to its chemical formula and osteoconductive nature, HA is resorbed and remodeled very slowly and could break under cyclic load before osteointegration is achieved [35]. Since patients do not benefit from bone substitutes in minimal defects, the impact of these biomaterials remains questionable for this indication [36,37]. As we know from bone biology, the relationship between the density of the HA/collagen structure and its stiffness is dependent on the specific locotypical function. In the ear, for example, a mineral content of more than 80% allows vibration and sound transmission, but not energy resorption, while in long bones a mineral content about 20% reduces weight and allows energy resorption [38,39,40]. Regarding our experiments, the applied scaffold might require further optimization of the HA/collagen composition; moreover, the potential biomechanical influence of the external fixator remains unclear without further interpretation.

### 3.3. Fluorescent Labeling of USSC

In the present study, fluorescent labeling of the USSC did not affect the survival and proliferation of cells in vitro as could be demonstrated in unchanged cell viability. It was shown that fluorescent labeling did not affect the capacity to form cartilage in vivo as could be demonstrated by comparing in vivo cartilage repair with labeled and unlabeled MSC. The quality of the cartilage retrieved was comparable when either labeled or unlabeled chondrocytes were injected, as evaluated by toluidine staining. The fluorescence was uniformly maintained in the cartilage implant obtained from the labeled cells [41] with a high density of labeled chondrocytes. 

### 3.4. External Fixation

The external fixation system used significantly exceeded the physiological circumference of the animal limb and represented a burden for the sheep. One animal had a complete dislocation of the fixateur and had to be euthanized. Although great care was taken to keep the pins sufficiently clean, several pin infections occurred throughout the healing period.

### 3.5. Limitations of the Study

The present study was a pilot study conducted with only a small sample size in order to examine the feasibility of the study design. In this case, necessary modifications were identified, such as the need for a larger defect size than 2 cm in order to have a critical size defect. The small sample size (one single animal left in the HA group) does not allow any possible conclusion to be reached. Additionally, the use of the external fixateur as stabilization system of the defect has to be reconsidered. Another limitation of the present study was the follow-up time. A follow-up time of six months was chosen but in regard to the biomechanical data, maximal stability could be seen already after three months (after four months in the control group), which was emphasized by the radiological results. A shorter follow-up time could have revealed a greater influence of the design parameters.

## 4. Materials and Methods 

### 4.1. Animal Model and Study Design

Due to their similarities in macrostructure and dimensions with human long bones, sheep bones were selected for the following experiments [42]. All procedures were approved by the national district government (8.87-50-10.34.08.033), and animal care was implemented in agreement with the guidelines for the care and use of laboratory animals. The international animal research reporting in vivo experiments (ARRIVE) guidelines were carefully complied with.

Six 12-to-18-month-old, female, non-pregnant Bentheimer sheep with 38 kg bodyweight were used in three different groups. Two animals were assigned to each group. Two animals had been delivered by caesarean section in order to be able to harvest their autologous cord blood cells beforehand (Figure 7). These animals were reared to maturity for 12 months before entering the experiment. A full-size, 2.0-cm, mid-diaphyseal tibial defect with the periosteum completely removed in sheep served as a bone defect (Figure 8). The bone fragments were stabilized by an external fixateur combined with a rigidity measuring device. 

In the test group, 2 × 10^7^ autologous USSC were seeded on four hydroxyapatite (HA) blocks which were inserted into the tibial bone defect of the right leg of the animals (Figure 7). In the HA-group, four HA blocks per defect were inserted without cells, while in the control group the tibial bone defect of the right leg was left without biomaterial and cells. 

Monthly monitoring of bone regeneration by radiography and biomechanical stability measurement was carried out and in addition also blood analysis for systemic response control. After six months, animals were sacrificed and the bone regeneration was evaluated histomorphometrically. The labeled USSC were tracked in the organs.

### 4.2. Characterization of Autologous USSC

Cord blood was harvested by needle aspiration of the placentomes immediately after birth. The mononuclear cell fraction (USSC) was cultivated and expanded as described previously [43]. For phenotypic characterization, adherent cells of the first passage were used [44]. Not all clusters of differentiation (CD) markers could be examined because of limited availability of commercial antibodies cross reacting with the sheep species. The typical markers for mesenchymal stroma cells (MSC) CD14, CD44, CD90, CD34, and CD45 were measured by flow cytometry (Cytomics FC 500 FACS: Beckman Coulter, Krefeld, Germany) with the respective monoclonal antibodies: mouse anti-human CD14-fluorescein-labeled (FITC) (clone TuK4; Invitrogen/ThermoFisher Scientific, Darmstadt, Germany), mouse anti-sheep CD44-FITC (clone 25.32, Bio-Rad, Muenchen, Germany), mouse anti-human CD90-PE (clone 5E10; Beckmann Coulter), mouse anti-sheep CD45-FITC (clone 1.11.32 Bio-Rad) and phyco-erythrine-conjugated (PE) mouse anti-human/sheep CD34 (clone SPM123, Novus Biologicals, R&D Systems, Wiesbaden, Germany).

USSCs were also tested in vitro regarding their mesenchymal multipotency (osteo-, chondro-, adipogenic) prior to transplantation using lineage-specific stimulation protocols as described before [43,45]. USCCs were kept frozen at −80 °C until autologous transplantation more than 12 months later.

### 4.3. Labeling of USSC for Homing and Cell Tracking

Since safety is a major concern for future clinical applications, USSC were labeled in order to investigate the homing of the transplanted cells. An ideal imaging modality for tracking therapeutic cells in patients requires the imaging tags to be non-toxic, biocompatible, and highly specific. We used PKH-26 (Sigma-Aldrich, Deisenhofen, Germany), a red fluorochrome (excitation 551 nm/emission 567 nm), incorporated into the cell membrane lipid bilayer by selective partitioning without interfering with the biological and proliferative activity of the cells.

Three weeks before surgery, USSC were thawed and expanded in 20% FCS. They were then converted into serum-free culture and concomitantly cultured in 10% autologous serum for six days. Two hours before autologous transplantation, USSCs were detached from culture flasks before labeling with the fluorescent dye PKH-26. The labeling process was performed according to the manufacturer’s protocol with a final PKH-26 concentration of 5 × 10^−6^ M [41]. In brief, 2 × 10^7^ cells were mixed with an equal volume of labeling solution containing PKH-26 in dilution buffer and incubated for four min at room temperature (RT). The reaction was stopped by adding a double volume of autologous serum. Cells were washed twice with PBS and suspended in autologous serum for transplantation.

The labeling procedure was controlled via flow cytometry including staining of vital and dead cells. To test whether the fluorescent labeling affected the survival and proliferation of cells in vitro, labeled USSC were cultivated and tested for their cell viability using the CellTiter-Glo^®^ luminescent cell viability assay (Promega, Mannheim, Germany) as described before [46]. This assay quantifies the ATP present, which signals the metabolically active cells. Luminescence, produced by the luciferase-catalyzed reaction of luciferin and ATP, was measured using a multilabel plate reader (VICTOR3™ PerkinElmer LAS, Rodgau-Jügesheim, Germany). Additionally, the USSC growth on the biomaterial was documented via SEM analysis as described before [47].

### 4.4. Biomaterials

Three dimensional scaffolds (0.5 × 1 × 1 cm) of bovine hydroxyapatite (Orthoss™, Geistlich Surgery, Wolhusen, Switzerland) were used as a carrier for USSCs (Figure 8c). The material has been in clinical use for bone defects and has been tested previously on human bone marrow cells for in vitro osteogenic potential [22,48,49]. 

The bone substitute material is a natural nanocrystalline carbonated hydroxyapatite Ca_10_(PO_4_)_6_(OH)_2_, made from highly purified bovine bone mineral. All organic components are removed by a specific stepwise low heat and a chemical treatment [50]. The material displays a topography and biofunctionality similar to human bone with a high porosity and specific surface area of 79.7 m^2^/g [50]. The HA contains bimodal pores with interconnective nanopores (10–20 nm pore size) and macropores (100–300 μm pore size). The nanopores contribute to the hydrophilic behavior, the capillarity and the wettability. Within the scaffold, the nanopores enable the uptake and retention of large volumes of blood and other fluids. The macropores can act as a conduit allowing movement and adhesion of bone building cells throughout the HA scaffolding and provide the space for blood vessel ingrowth [51]. The resistance to axial compression is 2.5 to 3 Mpa [50].

### 4.5. Biomaterial Insertion

Prior to use, all HA blocks were degassed and equilibrated in PBS. In the test group, a total suspension volume of 80 µL containing 5 × 10^6^ USSC cells of the third passage was added to each HA block, resulting in 2 × 10^7^ labeled cells per defect. An initial incubation period of 20 min at 37 °C was allowed for cellular adherence before the HA blocks were inserted into the tibial bone defect. In the HA group, the HA blocks did not receive any further treatment besides PBS equilibration before implantation. Figure 8 gives an overview of the surgical setting.

### 4.6. Animal Surgery

Premedication was given i.v. with ketamine (2 mg/kg body weight (BW), Ketavet, Pfizer, Berlin, Germany), thiopental (5 mg/kg BW, Thiopental-Inresa, Inresa, Freiburg, Germany) and medetomidine (0.005 mg/kg BW, Dormitor^®^, Pfizer, Karlsruhe, Germany). The right hind limb was shaved, disinfected and draped. Surgery was carried out in a lateral decubitus position under general anesthesia with nitrous oxide and 0.8–1.0% isoflurane under assisted ventilation. Pain control and antibiotic prophylaxis were carried out directly after surgery and for three days after surgery with carprofen (4.4 mg/kg BW s.c., Rimadyl^®^, Pfizer) with amoxicillin (6.4 mg/kg s.c., Synulox, BW, Pfizer). 

The defect was stabilized with a custom-made external, double, half-ring fixateur (Jlizarov External Fixation System, Smith and Nephews; Marl; Germany) with an integrated measuring device, allowing in vivo measurements of torsional stiffness from which the progress in bone healing can be estimated [52,53]. Six pins were used to stabilize the fixateur. In the middle of the tibia diaphysis, the periosteum was removed, and a standardized, 2.0-cm, full-thickness defect was created with an oscillating saw (Figure 8a,b). In the test-group, four cell-loaded scaffolds were placed in the defect zone while in the HA group only the biomaterial blocks were inserted before closure of the wound (Figure 8c,d). The control group did not receive any scaffold material. After the surgery, Bovi Concept^®^ (Albrecht GmbH, Aulendorf, Germany) was administered for stimulation of the rumen. The animals were kept separately for the first day after surgery. After that, they were housed indoors together.

### 4.7. Biomechanical Testing 

In order to obtain information about bone regeneration and functional outcome, a quantitative assessment of torsional in vivo stiffness of the regenerating bone was performed without the need for fixateur removal and the associated complications. For this, an external ring fixateur with a specialized device was constructed: the torsional stiffness meter was designed as a conventional external fixateur with a modified double half-ring allowing free torsional sliding (Figure 9). The measuring component was attached to the fixateur construct to measure the induced loads and displacements across the bone regenerate tested. The specially designed aluminum double half-ring was connected via a low-friction bearing allowing free rotation of the two half-rings relative to each other [54,55] (Figure 9a). During the measurement, applied forces and resulting displacements were continuously monitored with a linear variable differential transducer (LVDT) and a 500 N load cell (Figure 9b) (Hottinger Baldwin, Darmstadt, Germany). The sensor signals were processed by MGC-Lab software (Hottinger Baldwin) [54,55]. Biomechanical measurements were performed immediately after surgery and after one, two three, four, and five months. The parameters for a standard measurement were set to a maximum distance of 5 mm and torsion force up to 20 N.

### 4.8. Radiographic Analysis and Analysis of Systemic Response 

Standardized X-rays (mobile X-ray system with C-arm: Philips BV Pulsera REL 2.3, model 718095, Philips Electronics NL BV, Eindhoven, The Netherlands), of the operated tibia in two planes (anterior-posterior/lateral) were carried out preoperatively and one, two, three, four, five, and six months after surgery. At the same time intervals, systemic response (infection/inflammation/immunogenic reactions) was monitored by blood differential counts. Blood was obtained from punction of the vena cephalica antebrachii.

### 4.9. Histochemical Analysis

After six months, animals were euthanized with a lethal dose of pentobarbital-Na (Narcoren^®^, Merial GmbH, Hallbergmoos, Germany). Tibia specimens were carefully dissected and fixed in buffered formaldehyde for 14 days. After dehydration they were embedded in methylmetacrylate resin (Technovit 9100 neu, Hereaus Kulzer, Weinheim, Germany), which was polymerized at −4 °C for 48 h, as described earlier [54]. Sections were cut along the longitudinal axis with the EXAKT cut thin-ground technology (Exakt Apparatebau, Norderstedt, Germany) and ground to a final thickness of approximately 30 µm. The tibia sections were stained with toluidine blue (0.1% toluidine blue in 0.1% sodium tetraborate; Merck, Darmstadt, Germany) according to established protocols [56,57]. After dehydration, the slides were embedded in DePex (Merck).

For image acquisition, a color CCD camera (Color View III, Olympus, Hamburg, Germany) was mounted on a binocular light microscope (Olympus SZ 61, Olympus). The percentage of toluidine blue-stained new bone was measured in the defect area using the software ImageJ (1.48v, Java 1.8.0_191 (32bit), National Institutes of Health: http.//imagej.nih.gov/ij, USA). The area between the original bone fragments was identified and the bone area determined (Figure 10). One experienced investigator blinded to the specific experimental condition performed the histomorphometrical analysis.

### 4.10. Detection of Labeled USSC

Defined organs (spleen, kidney, lung, heart, liver, pancreas, duodenum and lymph nodes) were harvested by autopsy, embedded in Tissue-Tek^®^ O.C.T.™ (Sakura, Finetech Europe Zouterwoude, The Netherlands), and snap-frozen in liquid nitrogen. Eight-µm-thick cryosections through the center of each organ were used for direct observation of the PKH-26 labeled cells by fluorescent microscopy (Zeiss Axioscope 2, camera Axiocam MRc, software Axiovision v4.8, Carl Zeiss GmbH, Jena, Germany) or for histochemical examination by hematoxylin-eosin staining for macroscopic and microscopic evaluation in terms of potential tumor formation.

### 4.11. Data Analysis

Continuous variables are presented as mean ± standard deviation and categorical variables as frequency and percentage using SPSS 22.0 software (IBM Co., Armonk, NY, USA).

## 5. Conclusions

Although biomechanical stability was reached earliest in the test group with autologous USSC on HA scaffold, the density of newly formed bone was superior in the control group without any bovine HA.

## Figures and Tables

**Figure 1 molecules-24-00295-f001:**
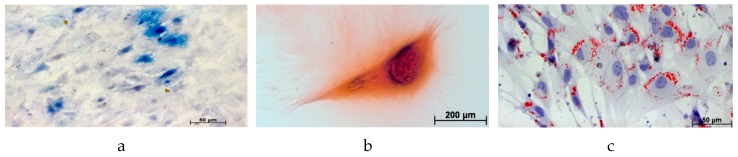
Differentiation of the USSC into the three lines. (**a**) osteogenic differentiation, alkaline phosphatase; (**b**) chondrogenic differentiation, safranin O; (**c**) adipogenic differentiation, oil red O.

**Figure 2 molecules-24-00295-f002:**
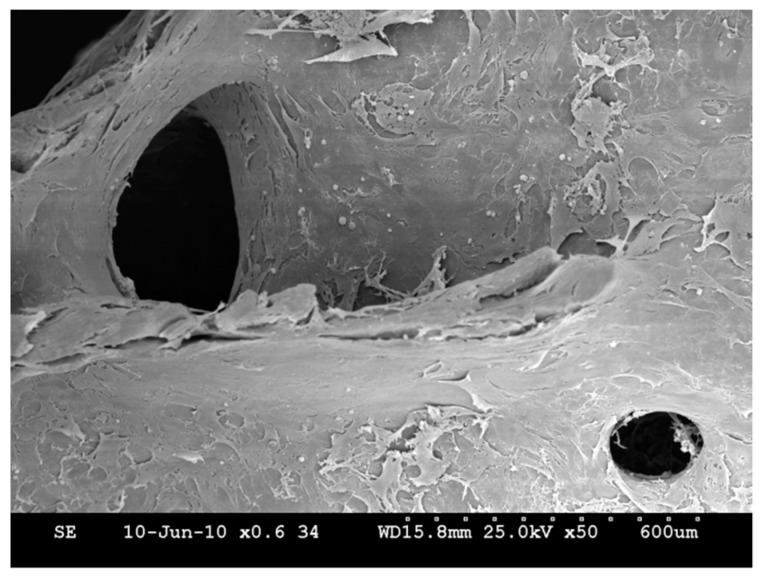
SEM picture of HA scaffold with USSC at day 10.

**Figure 3 molecules-24-00295-f003:**
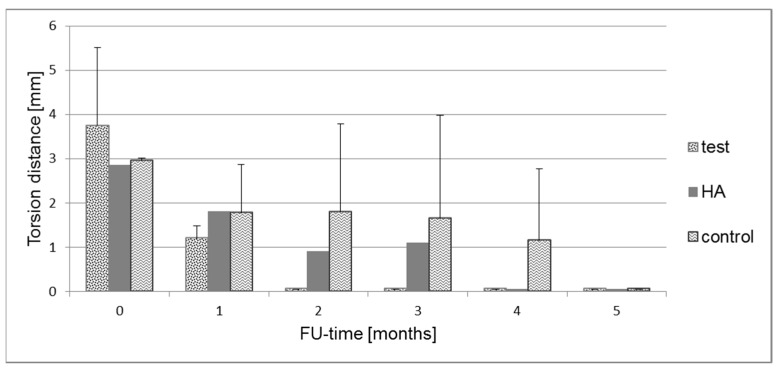
Biomechanical measurement showing the respective torsion distance until the torsion force of 20 N was reached.

**Figure 4 molecules-24-00295-f004:**
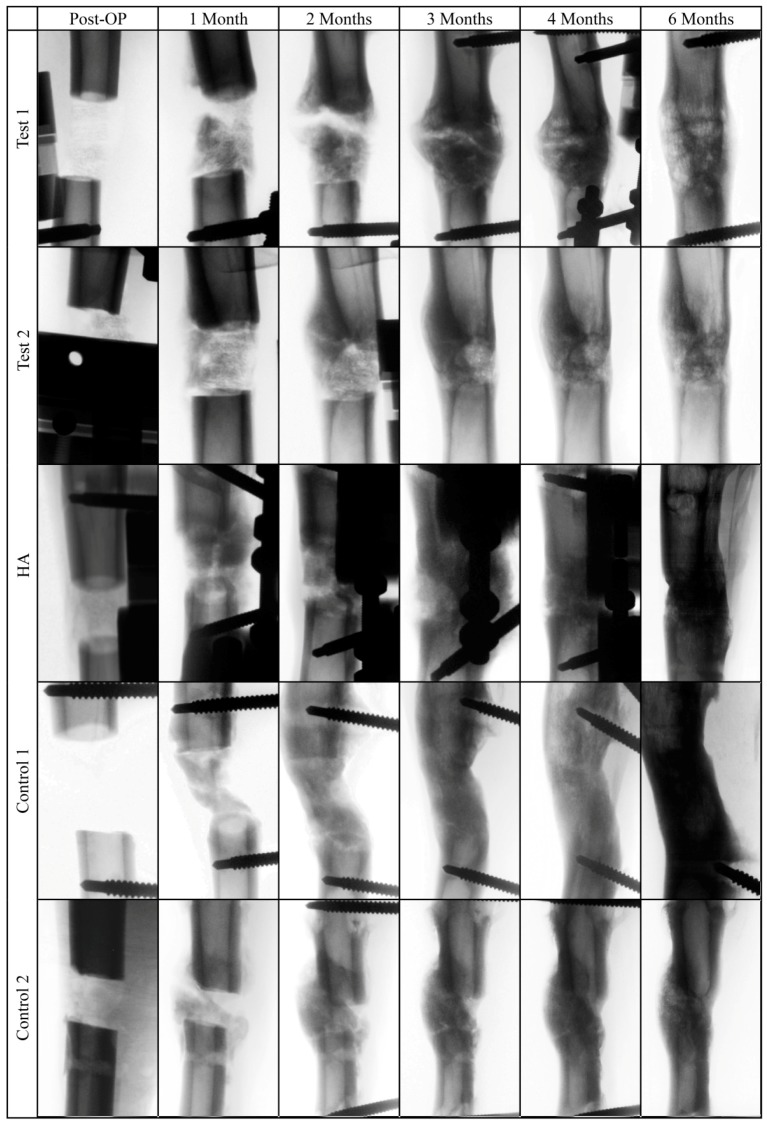
Representative invers radiograph images of the tibial bone defect area after surgery and after one, two, three, four, and six months.

**Figure 5 molecules-24-00295-f005:**
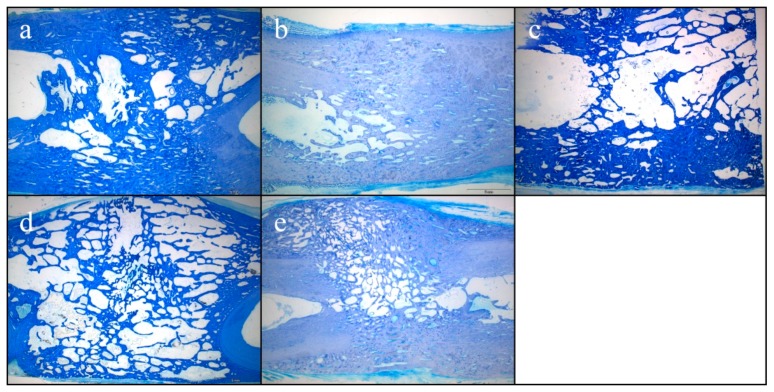
Toluidine blue staining of the former tibial bone defect area: (**a**,**d**) test group; (**b**,**e**) control group; (**c**) HA-group.

**Figure 6 molecules-24-00295-f006:**
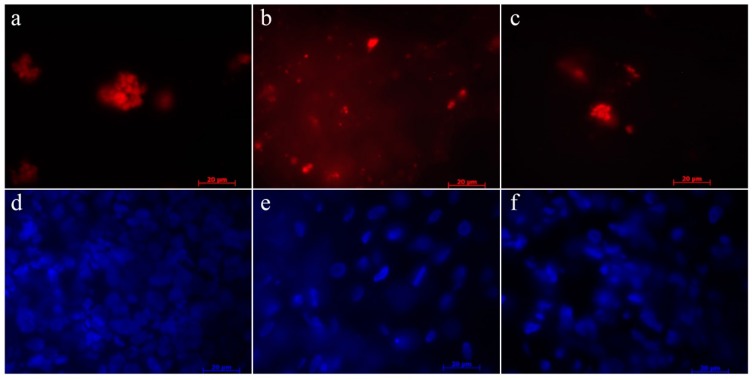
Homing of PKH-26 (red fluorescence) labeled MSC in (**a**) lymph nodes, (**b**) liver, and (**c**) pancreas of the animals of the test group. DAPI-staining of cell nuclei in (**d**) lymph nodes, (**e**) liver, and (**f**) pancreas.

**Figure 7 molecules-24-00295-f007:**
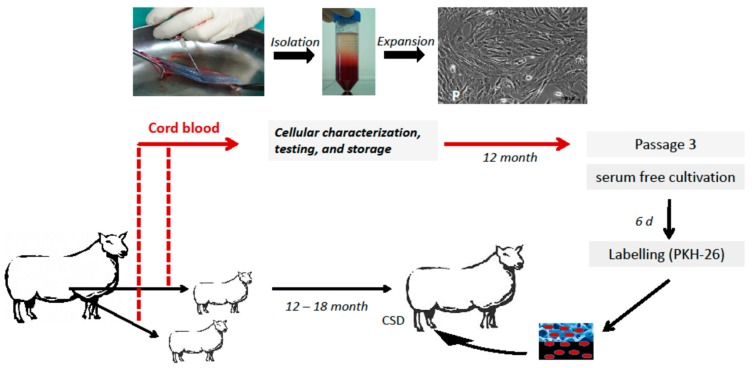
Experiment design of the test group: autologous implantation of labeled USSC into the tibial bone defect. Ovine USSC were obtained, isolated; expanded, characterized by flow cytometry, and frozen. After 12 months, full-size 2.0 cm mid diaphyseal bone defects were created in the right tibia of adult sheep. Autologous USSC of the third passage were thawed, labeled with the membrane dye PKH-26 and 2 × 10^7^ cells were seeded onto four hydroxyapatite (HA) blocks and implanted into the defect.

**Figure 8 molecules-24-00295-f008:**
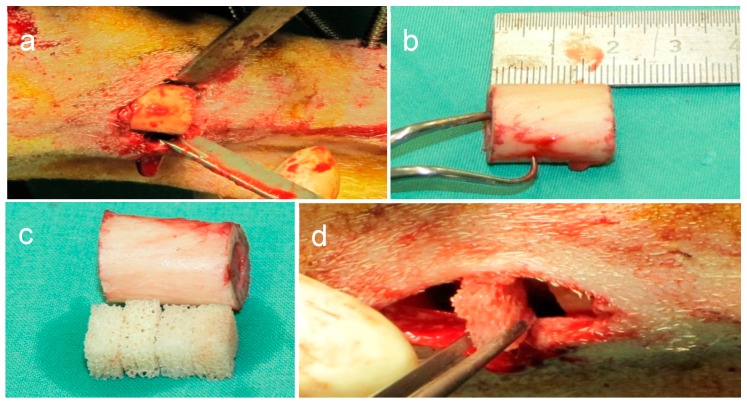
Bone defect. (**a**) In the middle of the tibia diaphysis the periosteum was removed and (**b**) a standardized 2.0 cm full thickness defect was created. (**c**) The bone fragment was replaced by four biomaterial scaffold blocks (test and HA group only), (**d**) which were placed in the defect area.

**Figure 9 molecules-24-00295-f009:**
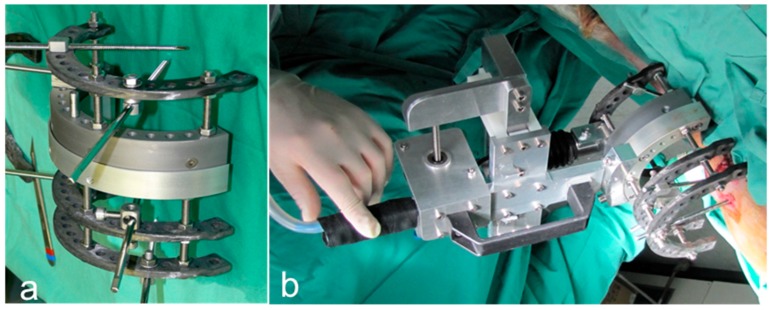
(**a**) Conventional external fixateur with a modified double half-ring allowing free torsional sliding. (**b**) Fixateur completely mounted with a torsional stiffness meter.

**Figure 10 molecules-24-00295-f010:**
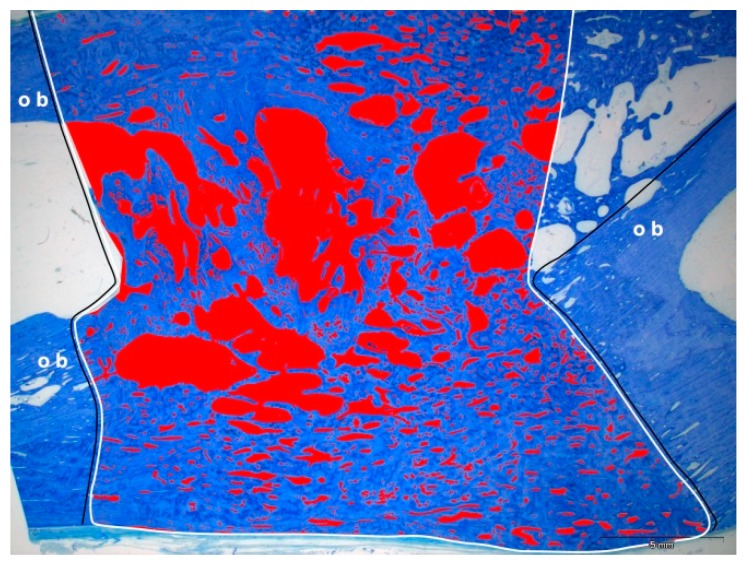
Schematic view of the histomorphometrical analysis. After identification of the original bone (ob), the defect area in between was determined. The area without bone (in red) was measured and subtracted from the total measurement area (in white), resulting in the area of newly formed bone.

**Table 1 molecules-24-00295-t001:** Histomorphometric analysis of the newly formed bone area (bone defect area) in between the original bone fragments.

Group	Distance between Original Bone Fragments (mm)	Measurement Area (mm^2^)	Bone Area (mm^2^)	Bone Area Percentage (%)
Test	22.7 ± 2.3	452.1 ± 55.4	262.5 ± 40.2	59.2 ± 13.0
HA	21.9	457.0 ± 08.8	222.0 ± 15.5	48.6 ± 2.9
Control	16.8 ± 0.8	298.2 ± 63.4	246.3 ± 57.0	82.5 ± 5.5

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
