# Peer review of "Biomechanical Stability and Osteogenesis in a Tibial Bone Defect Treated by Autologous Ovine Cord Blood Cells—A Pilot Study"

_molecules, 2019, doi:10.3390/molecules24020295_

Round 1

Reviewer 1 Report

In their study „Biomechanical stability and in vitro osteogenesis after CSD regeneration with autologous ovine cord blood cells - a pilot study” the authors address the timely and interesting topic of bone healing augmentation using stromal/stem cells (umbilical cord-derived cells) using an ovine tibial defect model. Although this study is of interest, there are several major issues that need to be addressed and it actually doesn’t fit with the scope of the journal “Molecules”.

The title doesn’t seem appropriate. Although the authors discuss that the model employed was not a critical size defect model, they claim to have evaluated the utility of autologous cord blood cells to regenerate a critical tibial defect. Also, within the manuscript they repeatedly speak of a critical defect.

Further, the term “in vitro osteogenesis” is puzzling, as they assessed osteogenesis in vivo and the study actually contains no meaningful in vitro data.

The final conclusion in the introduction section (line73/74) is incorrect, as no critical size defect was investigated!

The authors claim that the USSC seeded onto the scaffolds were mainly found on the scaffold surface and later also seemed to penetrate the scaffold (section 2.2).  However, the authors hardly provide any data on this and also provide no information on this in the methods section. How were the seeded scaffolds actually kept in culture (static or dynamic culture), how was viability tested (this is not possible via SEM)? The reviewer is surprised, that the study authors didn’t conduct for e.g. any histology on sections of the cultured scaffolds or performed a true live/dead assay.

Along the same lines, the authors claim to have assessed the multipotency of the isolated cells, but again do not show any actual data for this?

Point 3.4: The authors need to clearly state in which group the animal had to be euthanized. Based on the data shown in figure 2 the HA group seems to be affected. This is very problematic as this group then actually only contains a single animal. Even for a pilot study this is not acceptable and certainly does not allow any solid conclusions to be made.

Along the same lines, the statement that “…biomechanical stability was reached earliest in the test group with autologous USSC on HA scaffold…(line 346)” seems overemphasized. Actually, based on the radiograph shown in figure 3 (test group; 2 months) a clear gap is visible and I am very surprised that the torsion testing at 2 months suggests torsional stability?

Histology – it is a shame that only toluidine blue staining was performed on the PMMA sections. A more detailed histological analysis would be highly appreciated, including for example Movat's pentachrome stain and also the investigation if any of the fluorescently labelled cells actually directly contributed to the formation of bone tissue, or if this was mainly a paracrine effect.

Finally, statistical analysis on 2 biological replicates (or even a single animal) are by no means meaningful and allow no conclusions to be made.

Although the reviewer appreciates that the model used is a time consuming and expensive, the study overall is so flawed that practically no meaningful conclusions can be made.

Minor:

The authors confuse the terms totipotent, pluripotent, and multipotent (e.g. line 63.). By definition USSCs are at best pluripotent, but not totipotent. Actually, there is still quite some discussion if they genuinely contain a pool of pluripotent cells and generally they are believed to be multipotent.

The authors mix up the terms osteoinductive and osteoconductive (line 55). Growth factors (e.g. Bmp2) are osteoinductive.

Author Response

Dear Reviewer of the Manuscript “Biomechanical stability and in vitro osteogenesis after CSD regeneration with autologous ovine cord blood cells - a pilot study”

Thank you for taking the time and effort to work through the manuscript and the feedback for improving its quality. Within the next lines we will comment on your suggestion.

Point 1: The title doesn’t seem appropriate. Although the authors discuss that the model employed was not a critical size defect model, they claim to have evaluated the utility of autologous cord blood cells to regenerate a critical tibial defect. Also, within the manuscript they repeatedly speak of a critical defect.

Thank you for your indication of this disagreement. In the title, we omitted critical size defect model and changed it to “Biomechanical stability and osteogenesis in a tibial bone defect treated by autologous ovine cord blood cells - a pilot study”. Within the entire manuscript we replaced the term “critical size defect (CSD)” with “tibial bone defect” or only “bone defect”.

Point 2: Further, the term “in vitro osteogenesis” is puzzling, as they assessed osteogenesis in vivo and the study actually contains no meaningful in vitro data.

Thank you for pointing out this imprecision. We omitted the term “in vitro” in the title.

Point 3: The final conclusion in the introduction section (line73-74) is incorrect, as no critical size defect was investigated!

In last paragraph of the introduction, “we hypothesize that mesenchymal progenitor cells from the umbilical cord display an osteogenic differentiation potential in vivo and are capable of regenerating critical size osseous defects.” (line 78-79) This was our hypothesis when we started. It only became apparent during the implementation that the size of our bone defect (2 cm) was not a critical size defect. Therefore this is not our conclusion but our hypothesis which was in the end disproved.

In line 80, the term was changed to bone defect omitting “critical size”.

Point 4a: The authors claim that the USSC seeded onto the scaffolds were mainly found on the scaffold surface and later also seemed to penetrate the scaffold (section 2.2).  However, the authors hardly provide any data on this and also provide no information on this in the methods section.

Thank you very much for this point. We omit in this description …”mainly” (line 103). We also skipped the sentence: “At further follow-up, cell viability testing revealed that the cells completely covered the outer surface of the scaffold but were also detected at deeper levels up to 1 mm from the surface” because we do not present the corresponding cell viability data.

Point 4b: How were the seeded scaffolds actually kept in culture (static or dynamic culture), how was viability tested (this is not possible via SEM)? The reviewer is surprised, that the study authors didn’t conduct for e.g. any histology on sections of the cultured scaffolds or performed a true live/dead assay.

The USSC were added onto the HA scaffold immediately before insertion into the bone defect. Therefore the seeded scaffolds were not kept in culture apart from a short incubation time of 20 min. As stated in line 324 – 329, “Prior to use, all HA blocks were degassed and equilibrated in PBS. In the test group, a total suspension volume of 80 µl containing 5 x 106 USSC cells of the third passage was added to each HA block, resulting in 2 x 107 labelled cells per defect. An initial incubation period of 20 min at 37°C was allowed for cellular adherence before the HA blocks were inserted into the tibial bone defect.”

Beforehand, live / dead assays were done for the labeled cells as well as cell viability test via ATP assay as described in line 298-305.

Point 5: Along the same lines, the authors claim to have assessed the multipotency of the isolated cells, but again do not show any actual data for this?

Thank you for mentioning this Point. We described the characterization in the method section “Characterisation of autologous USSC” (line 261 – 273). In the first manuscript version we did not show any data since we wanted to focus on bone regeneration. In the revision we inserted a figure displaying the differentiation of the USSC into the three lines (new Fig. 1).

Point 6: The authors need to clearly state in which group the animal had to be euthanized. Based on the data shown in figure 2 the HA group seems to be affected. This is very problematic as this group then actually only contains a single animal. Even for a pilot study this is not acceptable and certainly does not allow any solid conclusions to be made.

It was stated in the results section 2.4 “In vivo assessment of bone healing “(line 117-119) that “The operated tibias were stably fixed with the external fixateurs except for one animal from the HA group, which had a complete dislocation of the fixateur and was euthanized 1 week after surgery.”

We are aware that due to the group size of one single animal no solid conclusion can be made. We emphasized this in the discussion section in the limitations by the insertion of the sentence: “The small sample size (one single animal left in the HA group) does not allow any possible conclusion to be reached” (line 221-223).

In accordance with the terms of “The ARRIVE Guidelines Checklist - Animal Research: Reporting In Vivo Experiments, we are obliged to present these data as discussion for the scientific community. Therefore, as recommended, we “comment on the study limitations including any potential sources of bias, any limitations of the animal model, the imprecision associated with the results, and any implications of your experimental methods or findings for the replacement, refinement or reduction (the 3Rs) of the use of animals in research.”
(Kilkenny C. et al, 2010, PLOS one; doi.org/10.1371/journal.pbio.1000412).

We hope that we answered your suggestion to your satisfaction and that you agree with our opinion.

Point 7: Along the same lines, the statement that “…biomechanical stability was reached earliest in the test group with autologous USSC on HA scaffold…(line 346)” seems overemphasized. Actually, based on the radiograph shown in figure 3 (test group; 2 months) a clear gap is visible and I am very surprised that the torsion testing at 2 months suggests torsional stability?

A torsional stability can go along with a clear gap in radiography since for a radiologic detection a certain degree of mineralization is necessary. Only the hard callus is visible in a radiographic picture.

https://www.researchgate.net/publication/285753966_Grundlagen_der_Frakturheilung_und_Bedeutung_fur_die_Osteosynthese

Point 8: Histology – it is a shame that only toluidine blue staining was performed on the PMMA sections. A more detailed histological analysis would be highly appreciated, including for example Movat's pentachrome stain and also the investigation if any of the fluorescently labelled cells actually directly contributed to the formation of bone tissue, or if this was mainly a paracrine effect.

Thank you for this aspect. Toluidine blue was chosen because it allows differentiation between old bone and new bone tissue for the histomorphometric analysis.

The fluorescently labelled cell can only be detected in only in frozen sections, which excludes bone tissue. We used methylmetacrylate resin (MMA, Technovit 9100 neu) for embedding of the bone because we did not want to demineralize the tissue beforehand. In the process of MMA embedding (and also for paraffin embedding), solvents like ethanol and acetone are used. Unfortunately, the membrane dye PKH 27 is not compatible with these solvents and cannot be displayed in MMA or paraffin embedded sections.

Unfortunately we could not transfect the USSC with GFP, which would have been the best way. Implantation of transfected cells requires a biological safety level (S1) for the animal holding facility, which was not available for housing of large animals.

So far I do not know of any paper showing that USSC or MSC directly contributed to the formation of bone tissue.

Point 9: Finally, statistical analysis on 2 biological replicates (or even a single animal) are by no means meaningful and allow no conclusions to be made.

Thank you for this point. In the method section, the subheading statistical analysis was omitted and changed to only “Data analysis” (line 427). In the following sentence, the term statistical analysis was omitted and changed to: Continuous variables are presented as mean ± standard deviation and categorical variables as frequency and percentage using SPSS 22.0 software (IBM Corp, Armonk, NY) (line 429-430).

Point 10: Although the reviewer appreciates that the model used is a time consuming and expensive, the study overall is so flawed that practically no meaningful conclusions can be made.

Minor:

Point 11: The authors confuse the terms totipotent, pluripotent, and multipotent (e.g. line 63.). By definition USSCs are at best pluripotent, but not totipotent. Actually, there is still quite some discussion if they genuinely contain a pool of pluripotent cells and generally they are believed to be multipotent.

Thank you for the update on the terminology. In line 64, we corrected the term from pluripotent to multipotent.

Point 12: The authors mix up the terms osteoinductive and osteoconductive (line 55). Growth factors (e.g. Bmp2) are osteoinductive.

Thank you for this point. In line 54-55, we changed the sentence to: …”are only osteoconductive and show a lack in osteoinductivity….

Reviewer 2 Report

1 Did the author test the compression or shear strength of the developed HA-based scaffolds?

2. Was the porosity of the samples optimized? Please provide more details than already available.

3. How significantly the Young modulus of the developed scaffolds differs from that of bone tissue used in experiments. Can we expect that no stress-shielding effect occurs in a long-range prospective.

4. HA-based scaffolds are used relatively long time. The novelty of the present study relative to the object for investigations should be specified in more details.

5. Why the authors have chosen 6 and 12 months in in vivo trials. Please provide more information on that.

Author Response

Dear Reviewer of the Manuscript “Biomechanical stability and in vitro osteogenesis after CSD regeneration with autologous ovine cord blood cells - a pilot study”

Thank you for taking the time and effort to work through the manuscript and the feedback for improving its quality. Within the next lines we will comment on your suggestion.

Point 1: Did the author test the compression or shear strength of the developed HA-based scaffolds?

Point 2: Was the porosity of the samples optimized? Please provide more details than already available.

Thank you for pointing out that no details on the scaffold material were given. We added a section about the scaffold material describing its structure, porosity and test of compression (line 312-321).

“The bone substitute material is a natural nanocrystalline carbonated hydroxyapatite Ca10 (PO4)6 (OH)2, made from highly purified bovine bone mineral. All organic components are removed by a specific stepwise low heat and a chemical treatment [50]. The material displays a topography and biofunctionality similar to human bone with a high porosity and specific surface area of 79.7 m2/g [50]. The HA contains bimodal pores with interconnective nanopores (10-20 nm pore size) and macropores (100-300 μm pore size). The nanopores contribute to the hydrophilic behavior, the capillarity and the wettability. Within the scaffold the nanopores enable the uptake and retention of large volumes of blood and other fluids. The macropores can act as a conduit allowing movement and adhesion of bone building cells throughout the HA scaffolding and provide the space for blood vessels ingrowth [51]. The resistance to axial compression is 2.5 to 3 Mpa [50].”

Point 3: How significantly the Young modulus of the developed scaffolds differs from that of bone tissue used in experiments. Can we expect that no stress-shielding effect occurs in a long-range prospective.

Unfortunately we do not have any information about the shear stress and Young modulus of the HA material used.

In the literature, the Young modulus of femur/tibia of cow is 18.49 ± 2.84 Gpa and comparable to sheep metacarpus with 18.95 ± 2.21 GPa. (Currey, JD. The effect of porosity and mineral content on the Young’s modulus of elasticity of compact bone. J. Biomechanics Vol. 21, No. 2. pp 131-139. 1988).

We would expect that this might be also valid for the scaffold material from bovine bone, since there is no effect of preservation on the young modulus. (Unger S, Blauth M, Schmoelz W. Effects of three different preservation methods on the mechanical properties of human and bovine cortical bone. Bone Volume 47, Issue 6, pp.1048-1053).

Concerning the stress shielding effect, the use of the fixateur extern will remove the typical stress from the bone. A stress-shielding effect is expected since the bone will remodel in response to the loads it is placed under.

Concerning the stress shielding effect, the use of the fixateur extern will remove the typical stress from the bone. A stress-shielding effect is expected since the bone will remodel in response to the loads it is placed under.

Point 4: HA-based scaffolds are used relatively long time. The novelty of the present study relative to the object for investigations should be specified in more details.

Thank you for indicating this important issue. We added a sentence concluding the novelty of the present study in the introduction (line 81-83). “The novelty of the present study is the use of autologous USSC for bone repair with the concurrent monitoring of bone regeneration by radiography and biomechanical stability.”

Point 5: Why the authors have chosen 6 and 12 months in in vivo trials. Please provide more information on that.

We discovered that we did not clearly defined the animal rearing time and follow-up periods in the experiment design.

The experiment starts with the birth of the sheep and the harvest the umbilical cord blood. The USSC are isolated, cultivated, characterized and frozen. The animals are reared for at least 12 months to be old enough to enter the surgery procedure in which the animals of the test group receive their own cord blood cells back on HA scaffold into their bone defect (autologous approach). After surgery, bone regeneration was monitored by radiography and biomechanical stability measurements were carried out and in addition to blood analysis (systemic response control) on a monthly base. After 6 months, animals were sacrificed and the bone regeneration was evaluated histomorphometrically.

We added this information to the study protocol (line 248 – 251).

We have chosen 12 months for the new born sheep to reach maturity.
The follow up time after creation of the bone defect was chosen to be 6 months because this is the maximum time for a complete fracture healing. If there is still a fracture gap after this time, the healing is incomplete and defined as nonunion.

Reviewer 3 Report

Presented articel is interesting and presents valuable for clinicians informations. The animal model is well done. It is very pityy, the Authors did not made micrCT to evaluation of bone dinsity. The Authors did not evaluated new vascularisation using specyphic stainnings on imunohistological slides. It should be done. In introduction please develop the section about HA. It should be stress more data about that source of materials i. (https://journals.sagepub.com/doi/abs/10.4137/BTRI.S36138), (https://www.ncbi.nlm.nih.gov/pmc/articles/PMC5461442/) , (https://www.ncbi.nlm.nih.gov/pubmed/28575969), and (https://www.ncbi.nlm.nih.gov/pubmed/27612684)

Author Response

Dear Reviewer of the Manuscript “Biomechanical stability and in vitro osteogenesis after CSD regeneration with autologous ovine cord blood cells - a pilot study”

Thank you for taking the time and effort to work through the manuscript and the feedback for improving its quality. Within the next lines we will comment on your suggestion.

Point 1: The Authors did not evaluated new vascularisation using specific stainings on immunohistological slides. It should be done.

Thank you for mentioning this additional point of view on the bone regeneration. We did immunohistology for osteocalcin and for transglutaminase II, a marker for endothelial cells. Pictures could be inserted. No quantitative evaluation of vascularization was done.

Point 2: In introduction please develop the section about HA. It should be stress more data about that source of materials.

We added a section about the scaffold material describing its structure, porosity, and test of compression (line 312-321).